# Polygalae Radix Oligosaccharide Esters May Relieve Depressive-like Behavior in Rats with Chronic Unpredictable Mild Stress via Modulation of Gut Microbiota

**DOI:** 10.3390/ijms241813877

**Published:** 2023-09-09

**Authors:** Qijun Chen, Tanrong Jia, Xia Wu, Xiaoqing Chen, Jiajia Wang, Yinying Ba

**Affiliations:** 1School of Traditional Chinese Medicine, Capital Medical University, No. 10, Xitoutiao, Youanmenwai Street, Beijing 100069, China; qijun@ccmu.edu.cn (Q.C.); jiatanrong@126.com (T.J.); wuxia6710@163.com (X.W.); cxq1983716@163.com (X.C.); wjj8935@163.com (J.W.); 2School of Pharmaceutical Sciences, Capital Medical University, No. 10, Xitoutiao, Youanmenwai Street, Beijing 100069, China

**Keywords:** polygalae radix oligosaccharide esters, depression, chronic unpredictable mild stress, gut microbiota, intestinal barrier

## Abstract

Polygalae radix (PR) is a well-known traditional Chinese medicine that is used to treat depression, and polygalae radix oligosaccharide esters (PROEs) are the main active ingredient. Although gut microbiota are now believed to play key role in depression, the effects of PROEs on depression via modulation of gut microbiota remain unknown. In this article, we investigate the effect of PROEs on the gut microbiota of a depression rat and the possible mechanism responsible. The depression rat model was induced by solitary rearing combined with chronic unpredictable mild stress (CUMS). The depression-like behavior, the influence on the hypothalamic–pituitary–adrenal (HPA) axis, the contents of monoamine neurotransmitter in the hippocampus, and the quantity of short-chain fatty acids (SCFAs) in the feces were each assessed, and the serum levels of lipopolysaccharide (LPS) and interleukin-6 (IL-6) were measured by ELISA. Additionally, ultrastructural changes of the duodenal and colonic epithelium were observed under transmission electron microscope, and the gut microbiota were profiled by using 16S rRNA sequencing. The results show that PROEs alleviated the depression-like behavior of the depression model rats, increased the level of monoamine neurotransmitters in the brain, and reduced the hyperfunction of the HPA axis. Furthermore, PROEs regulated the imbalance of the gut microbiota in the rats, relieving intestinal mucosal damage by increasing the relative abundance of gut microbiota with intestinal barrier protective functions, and adjusting the level of SCFAs in the feces, as well as the serum levels of LPS and IL-6. Thus, we find that PROEs had an antidepressant effect through the restructuring of gut microbiota that restored the function of the intestinal barrier, reduced the release of intestinal endotoxin, and constrained the inflammatory response.

## 1. Introduction

As a mood disorder disease, depression is characterized by anhedonia, unresponsiveness, poor appetite, hypokinesia, insomnia, and suicidal tendencies [1], and according to the World Health Organization, depression will be the largest contributor to worldwide disease burden by 2030 [2]. The pathogenesis of depression is complex and is usually associated with dysregulation of the hypothalamic–pituitary–adrenal (HPA) axis, monoaminergic deficiencies, immune system dysfunction, and gut microbiota disorders [3]. Recent studies have demonstrated that gut microbiota and their metabolites in particular play an important role in the pathogenesis and treatment of depression.

The gut microbiome includes all the microorganisms that inhabit the gut and their respective genomes [4]. Gut microbiota regulate a variety of host metabolic pathways, including intestinal motility, intestinal barrier homeostasis, nutrient absorption, and fat distribution [5]. The interaction between gut microbiota and the intestine is also involved in the regulation of nervous system function, which affects the occurrence and development of various diseases, including both gastrointestinal diseases and psychiatric diseases [6]. Changes in gut microbiota can affect the central nervous system through the HPA axis, immune response, and neurotransmitter signal transduction, which can all participate in the occurrence and development of depression [3]. Numerous studies have also shown that there are significant differences in the compositions of gut microbiota between depressed patients and healthy individuals [7]. For example, germ-free mice inoculated with microbiota derived from depressed patients or healthy individuals exhibited depression-like or normal behaviors, respectively, and probiotic treatments have been shown to mitigate depression-like behavior in mice [8,9]. Similarly, gut-microbiota composition tends to normalize in patients with depression after efficaciously responding to antidepressants [7].

Polygalae radix (PR), the dried root of *Polygala tenuifolia* Willd. or *Polygala sibiria* L., is a well-known traditional Chinese medicine that has the effect of tranquilizing the mind, improving intelligence, and restoring normal coordination between the heart and kidneys, and this medicine is widely used for the treatment of insomnia, dreaminess, palpitations, and neurasthenia in clinic [10] in the form of Kai-Xin-San, Yuan-Zhi decoction, or Anshendingzhi decoction [11,12,13]. Modern pharmacological studies have also shown that PR has beneficial antidepressant, sedative, anti-aging, neuroprotective, intelligence, and immune system effects [14]. Polygalae radix oligosaccharide esters (PROEs) are the active ingredients in PR thought to be responsible for its antidepressive effects. Researchers have reported that 3,6′-dicalicosyl sucrose and tenuifoliside A are the main components of PR total oligosaccharide esters and that these compounds regulate the level of monoamine neurotransmitters and improve the function of the HPA axis [15,16,17]. Although studies have found that PROEs have poor absorption and low bioavailability, suggesting that they may exert an antidepressant effect via modulation of gut microbiota [18,19]. In this study, a rat depression model was established by feeding alone combined with chronic unpredictable mild stress (CUMS) to evaluate the effects of PROEs on their gut microbiota and to investigate the PROEs’ possible antidepressant mechanisms.

## 2. Results

### 2.1. Identification of Chemical Components in the PROEs

We analyzed the PROEs by UPLC-QTOF-MS, and the results showed that they contained mostly oligosaccharide esters. In total, we characterized 27 oligosaccharide esters from the PROEs in accordance with the existing literature [20,21,22] (Appendix A).

### 2.2. The Effect of PROEs on Depression Rat

To evaluate the antidepressant activity of PROEs, we recorded the body weights of the rats and performed the sucrose preference test (SPT), open-field test (OFT), and forced swimming test (FST) on each group. The CUMS challenge resulted in significant body-weight loss, but PROE treatment prevented this CUMS-induced weight loss (Figure 1A). As presented in Figure 1B–F, the sucrose preference rate was significantly lower in the CUMS group compared to the NC group (*p* < 0.001), and the sucrose preference rate was significantly higher in rats that were treated with FXT and PROEs for eight weeks as compared to the CUMS group (*p* < 0.01). In the FST, the rats in the CUMS group showed a significantly higher immobility time compared to the NC group (*p* < 0.001), but the FXT and PROE treatment groups had significantly reduced immobility times (*p* < 0.01). The total movement distance, number of rearings, and the time spent in the central area in the OFT were also all significantly lower in the CUMS group compared to the NC group (*p* < 0.01) but were significantly higher in the FXT and PROE groups as compared to the CUMS group (*p* < 0.01). These data suggest that the PROEs may have relieved the CUMS-induced depression-like behavior.

### 2.3. The Effect of PROEs on Monoamine Neurotransmitters and Their Metabolites in the Hippocampus

As shown in Table 1, the content of 5-HT, NE, DA, DOPAC, 5-HIAA, and HVA were significantly lower in the hippocampus of CUMS rats compared to the NC group (*p* < 0.01), but the content of 5-HT, NE, DOPAC, and 5-HIAA were significantly higher in the FXT and PROE groups compared to the CUMS group (*p* < 0.05). Furthermore, compared to the CUMS group, the content of DA was significantly higher in the FXT and PROE-H groups (*p* < 0.01), and the content of HVA was significantly higher in the FXT, PROE-H, and PROE-M groups (*p* < 0.01).

### 2.4. The Effect of PROEs on HPA Axis

As shown in Figure 2A–C, the levels of CORT, ACTH, and CRF were significantly higher in the CUMS group compared to the NC group (*p* < 0.001), and the serum levels of CORT and ACTH were significantly lower in the FXT and PROE groups compared to the CUMS group (*p* < 0.001). In addition, the serum levels of CRF showed a downward trend in the PROE groups.

### 2.5. The Effects of PROEs on Gut Microbiota Composition

#### 2.5.1. Changes in Gut Microbiota Diversity

As shown in Figure 3A,B, the Shannon and Chao 1 index suggests that the CUMS challenge led to a significant increase in gut microbiota diversity, but the PROE-H groups had significant reductions in gut microbiota diversity compared to the CUMS group (*p* < 0.01). In addition, our alpha diversity analysis indicated that there was a significantly higher amount of richness and diversity in the gut microbiota of the CUMS group compared to the NC and PROE groups. Beta diversity analysis, however, usually begins by calculating the distance matrix to determine differences in overall microbial composition between samples. To this end, principal co-ordinate analysis (PCoA) showed that the structure of the gut microbiota in the CUMS group was significantly separated from that in the NC and NC-PROE groups, whereas the structure of the gut microbiota in the PROE group deviated from the CUMS group and approached that of the NC group (Figure 3C). Furthermore, the unweighted pair group method with arithmetic mean (UPGMA) also yielded the same results, indicating that PROEs changed the gut microbiota structure of the host (Figure 3D).

#### 2.5.2. Changes in the Composition of Gut Microbiota

Taxonomical analysis of microbial structure at the phylum level showed that Firmicutes and Bacteroidetes were the main phyla in all groups. Compared to the NC group, the relative abundance of the Firmicutes and actinobacteria of the CUMS group was significantly decreased (*p* < 0.01), and the relative abundances of Bacteroidetes and proteobacteria were significantly elevated, but PROE treatment restored the relative abundance of the microbiota (Figure 4).

At the family level (Figure 5), redundancy analysis (RDA) indicated that the CUMS group was clearly separated from the NC group, and the PROE-H group was close to the NC group. Bacteroidaceae, Muribaculaceae, and Lachnospiraceae were most affected by the CUMS protocol, whereas Peptostreptococcaceae, Ruminococcaceae, Lachnospiraceae, and Prevotellaceae were the dominant families in the NC and PROE-H groups. Additionally, the relative abundances of Bacteroidaceae and Muribaculaceae were significantly improved (*p* < 0.05), and those of Peptostreptococcaceae, Ruminococcaceae, and Lachnospiraceae were strikingly decreased in the CUMS group compared to the NC group (*p* < 0.05). Interestingly, the PROE treatment restored the relative abundances of the above gut microbiota at the family level.

At the genus level (Figure 6), the relative abundances of *Bacteroides*, *Oscillibacter*, *Parasutterella*, and *Intestinimonas* were significantly higher here (*p* < 0.05), but those of *Romboutsia*, *Roseburia*, *Lachnospiraceae_NK4A136_group*, *Prevotella_9*, and *Eubacterium_coprostanoligenes_group* were significantly lower in the CUMS group compared to the NC group (*p* < 0.05). However, once again, the PROE-H treatment restored the composition of the above gut microbiota at the genus level.

#### 2.5.3. Taxonomic Biomarkers in Rat Gut Microbiota

Linear discriminant analysis of effect size (Lefse) showed that there were 16 identified taxonomic biomarkers found with *p*-values < 0.01 and LDA scores (log 10) > 4.0, among which four were enriched in the CUMS group and three were enriched in the PROE-H group (Figure 7A). The four biomarkers in the CUMS group were all from Bacteroidetes and included Bacteroidaceae, Muribaculaceae, *Bacteroides_acidifaciens*, and *Bacteroides*. The three biomarkers enriched in the PROE-H group were from Firmicutes and Bacteroidetes and included *Prevotella_9*, Peptostreptococcaceae, and *Romboutsia*.

#### 2.5.4. Prediction of Metagenomic Functions

The KEGG pathways at the level-2 results showed that there were significant differences in terms of the nervous system, amino acid metabolism, immune system diseases, and neurodegenerative disease pathways between the CUMS group and NC group (*p* < 0.05), which illustrates that gut microbiota may reflect the physiological state of their host. Compared to the CUMS group, PROEs had significant differences in nervous system, amino acid metabolism, lipid metabolism, immune system diseases, and neurodegenerative disease pathways (*p* < 0.05). In addition, KEGG pathways at level 3 showed that the PROE-H group regulated depression-like pathophysiology by affecting amino acid metabolism and lipid metabolism, including tryptophan metabolism, valine/leucine/isoleucine degradation, glycine/serine/threonine metabolism, and sphingolipid metabolism (Figure 8).

### 2.6. Correlation Analysis among Brain Neurotransmitters, HPA Axis Hormone, and Relative Abundances of Gut Microbiota

Spearman’s correlation analysis revealed strong correlations among brain neurotransmitters, the HPA axis hormone, and specific gut microbiota genera (Figure 7B; |r| > 3, *p* < 0.05). Specifically, the content of 5-HT, DA, NE, HVA, DOPAC, and 5-HIAA were strongly negatively correlated with the genera *Bacteroides*, *Christensenellaceae_R-7_group*, *Lactobacillus*, *Parasutterella*, *Intestinimona*, and *Prevotellaceae_UCG-001* but positively correlated with the genera *Lachnospiraceae_NK4A136_group*, *Prevotella_9*, *Eubacterium_coprostanoligenes_ group*, *Prevotellaceae_NK3B31_group*, and *Romboutsia*. Additionally, the serum levels of CORT and ACTH were strongly negatively correlated with the genera *Romboutsia*, *Roseburia*, and *Ruminiclostridium_6* but were positively correlated with *Parasutterella* and *Oscillibacter*.

### 2.7. The Effect of PROEs on Plasma Tryptophan and Kynurenine Levels 

Compared to the NC group, the levels of TRP were significantly lower (*p* < 0.001), and the levels of KYN and the TRP/KYN ratio were significantly higher in the CUMS group (*p* < 0.01, Figure 2D–F). In the FXT and PROE groups, the levels of TRP were significantly higher than in the CUMS group (*p* < 0.01), but the levels of KYN and the TRP/KYN ratio were significantly lower than in the CUMS group (*p* < 0.05). These results demonstrate that PROEs may regulate the TRP-KYN metabolic pathway.

### 2.8. The Effects of PROEs on the Morphological Structure of the Intestinal Mucosa

#### 2.8.1. The Effects of PROEs on Duodenum and Colon Histopathological Changes

The results of our histopathological examination show that the duodenum and colon had intact mucosa, clear crypt (C), and villus (V) structure with adequate goblet cells in the NC group (Figure 9). The duodenal villi were severely shedding in depression-model rats, however, and the duodenal and colonic crypts were distorted, with massive inflammatory cell infiltration in the mucosa. However, PROE administration reversed these histopathological changes. Compared to the NC group, the length of the duodenal villi was significantly lower in the CUMS group (*p* < 0.001), the colonic villi had a downward trend, the depth of duodenal and colonic crypts were significantly higher (*p* < 0.001), and the ratio of V/C was significantly lower (*p* < 0.01). Furthermore, compared to the CUMS group, the length of the duodenal villi was significantly lower in the FXT and PROE-H groups (*p* < 0.001); these colonic villi had an upward trend, the depth of duodenal and colonic crypts were significantly lower (*p* < 0.001), and the ratio of V/C was significantly higher (*p* < 0.01).

#### 2.8.2. Effects of PROEs on Duodenum and Colon Epithelium Ultrastructure

In the NC group, the microvilli in the duodenum and colon were rich and orderly, and the structure was complete, with normal intercellular tight junctions, as observed by electron microscope. However, in the CUMS group, the duodenum and colon mucosa were seriously damaged, the microvilli were atrophic and even deficient; gaps in intercellular tight junctions became significantly larger, and the goblet cells had disappeared almost entirely. The intestinal mucosa injury of the FXT and PROE-H groups was improved, and the microvilli of the duodenum and colon were dense and arranged in an orderly manner. Moreover, the structure was complete, and the tight junctions returned to normal after treatment (Figure 10).

#### 2.8.3. Effects of PROEs on the Protein Expression of Occludin in the Colon

The protein expression of occludin in the colon of the CUMS group was significantly lower compared to the NC group (*p* < 0.05). Compared to the CUMS group, the protein expression of occludin was significantly higher in the PROE groups (*p* < 0.05), and this expression in the FXT group had an upward trend (Figure 11).

### 2.9. The Effect of PROEs on Serum IL-6 and LPS Levels 

As presented in Figure 2G,H, the levels of IL-6 and LPS were significantly higher in the CUMS group compared to the NC group (*p* < 0.001) but were significantly lower in the FXT and PROE groups as compared to the CUMS group (*p* < 0.05).

### 2.10. The Effect of PROEs on the Expression of 5-HT_1A_, 5-HT_2A_, IDO1, and TNF-α mRNA

As shown in Figure 12A–F, compared to the CUMS group, the expression of 5-HT_2A_ mRNA was significantly lower in the FXT and PROE groups (*p* < 0.01), and the expression of 5-HT_1A_ mRNA was significantly elevated in the FXT, PROE-M, and PROE-L groups (*p* < 0.05). The expression of IDO1 and TNF-α mRNA in the cerebral cortex and duodenum was significantly higher in the CUMS group compared to the NC group (*p* < 0.01), however, and the expression of IDO1 and TNF-α mRNA in the cerebral cortex and duodenum was significantly lower in the PROE groups compared to the CUMS group (*p* < 0.05).

### 2.11. The Effect of PROEs on the Concentrations of SCFAs in Feces

As shown in Figure 12G–I, the concentrations of acetic acid, propionic acid, and butyric acid in feces were significantly lower in the CUMS group than in the NC group (*p* < 0.01). Compared to the CUMS group, the concentrations of acetic acid and propionic acid were significantly higher in the FXT, PROE-H, and PROE-L groups (*p* < 0.05), and the concentrations of butyric acid were significantly higher in the PROE-H group (*p* < 0.05).

## 3. Discussion

Depression is a common and recurrent mood disorder illness. Although its pathogenesis is not completely clear, many studies have suggested that stress factors play a key role and that long-term stress is the main cause of depression. In this study, a depression rat model was established using solitary feeding combined with CUMS. CUMS rats were randomly given the stimulation of variable and unpredictable stress factors, which is similar to the hypothesized pathogenic process of human depression [23]. The body weights of the rats in the CUMS group decreased, and they showed depression-like behaviors such as loss of pleasure, increased demonstration of hopeless behavior, decreased autonomous activity, and decreased desire to explore new environments. Furthermore, compared to the NC group, the levels of DA, 5-HT, NE, DOPAC, and 5-HIAA in the hippocampi of the CUMS group were significantly lower, and the levels of CORT, ACTH, and CRF in serum were significantly higher. PROEs can evidently improve depression-like behaviors in model rats and the hyperfunction of the HPA axis, increasing the levels of monoamine neurotransmitters in the hippocampus, which indicates that PROEs may have an antidepressant effect.

In addition, we found that PROEs can regulate the gut microbiota of depression-model rats. In the analysis of gut microbiota structure, alpha diversity is a comprehensive indicator of richness and evenness that is commonly used to characterize the presence of different species in community ecology. Some previous studies have suggested that depression is associated with an increased richness and diversity of gut microbiota [24]. In this study, the difference of gut microbiota among different groups was directly observed by visualization methods in beta diversity analysis. This analysis showed that the structure of gut microbiota in the CUMS group was significantly changed but that PROEs made the structure of the gut microbiota closer to that of the NC group, which is consistent with the literature.

Studies have shown that there are significant changes in the species composition of gut microbiota in patients with depression and depression-model animals. At the phylum level, we found that PROEs significantly increased the abundance of Actinobacteria and Firmicutes and decreased that of Bacteroidetes, an important result given that the members of the phylum Actinobacteria might be potential biomarkers for ketamine’s antidepressant efficacy [25,26]. At the family and genus levels, we found that the gut microbiota regulated by PROE were mainly related to the protection of the intestinal barrier, and the production of SCFAs and endotoxins. Researchers have reported that various harmful bacteria, toxins, and some inflammatory factors induced by gut microbiota disorder can alter the integrity and permeability of tight connections and destroy the function of the intestinal mucosa barrier in different ways [27]. Researchers have also found that the structure of intestinal microvilli and epithelial in CUMS depression-model rats is seriously damaged, that their intestinal glands are reduced or even gone completely, and that there is inflammatory cell infiltration in the lamina propria mucosa [28]. Tight junctions are an important structure of the intestinal mucosal barrier, and decreased tight junction protein occludin expression leads to intestinal mucosal barrier dysfunction and alters intestinal permeability [29]. SCFAs such as acetic acid, propionic acid, and butyric acid, which are important metabolites of gut microbiota, can directly affect the permeability of intestinal mucosa. SCFAs are an important energy source for epithelial cells, which can stimulate cell proliferation [30]. Butyric acid, for example, has been shown to reduce diarrhea caused by intestinal barrier dysfunction, such as inflammatory bowel disease, as well as to increase the levels of occludin and cingulin proteins in HeLa cells [31].

In this study, PROEs significantly increased the abundance of both Prevotellaceae and Peptostreptococcaceae, which can ferment carbohydrates in food to produce SCFAs. A decrease in the abundance of these organisms has been shown to be related to increased intestinal permeability [32,33] in depression-model rats. At the genus level, the abundance of *Romboutsia*, *Roseburia*, *Lachnospiraceae_NK4A136_group*, *Prevotella_9*, and *Eubacterium_coprostanoligenes_group* was significantly lower in the CUMS group compared to the NC group, which has already been reported to be related to the maintenance of intestinal barrier function and the alleviation of inflammatory response. *Romboutsia*, *Roseburia*, *Lachnospiraceae_NK4A136_group*, and *Prevotella_9* can produce SCFAs [34,35,36,37], and the intestinal epithelial oxidative and inflammatory damage of mice rich in these four bacteria has been found to be lower than that of control mice [38]. *Lachnospiraceae_NK4A136_group* has also been found to be positively correlated with factors related to the maintenance of intestinal barrier integrity and negatively correlated with pro-inflammatory factors (LPS and IL-6) and neurotoxic quinolone [39,40]. In addition, the increased abundance of *Romboutsia* is associated with decreased levels of proinflammatory cytokines in plasma [34], and the lower the abundance of *Eubacterium_coprostanoligenes_group*, the more easily the intestinal barrier will be destroyed [41]. Our results also show that PROEs not only increased the abundance of beneficial bacteria but also reduced the relative abundance of harmful bacteria such as *Bacteroides Oscillibacter*, *Parasutterella*, and *Intestinimonas* in the depression-model rats. Studies have reported that the relative abundance of *Oscillibacter* and *Parasutterella* is higher in cases of major depressive and bipolar disorder [42,43]. *Bacteroides* are a kind of Gram-negative bacteria that increase the production of harmful metabolites such as amyloids, LPS, enterotoxins, and neurotoxins [44]. In addition, LPS and other bacterial endotoxins are more easily absorbed into the blood after the intestinal barrier function is damaged, and this can induce the release of peripheral and central proinflammatory cytokines. Proinflammatory cytokines play an important role in the pathogenesis of depression, and the levels of proinflammatory cytokines in depression patients are generally higher than those of a normal person [45].

In addition to mediating the microbiota, we also observed that PROEs relieved intestinal mucosal damage, increased the expression of occludin protein in the colon, adjusted the level of SCFAs in the feces and serum levels of LPS and IL-6, and reduced the expression of TNF-α m RNA in the cerebral cortex and duodenum. One study has reported that RP can inhibit the activation of NLRP3 inflammasome and the production of pro-inflammatory cytokines such as TNF-α in the prefrontal cortex of rats, thereby exerting antidepressant effects by promoting autophagy and inhibiting neuroinflammation [46]. Our results indicate that PROEs may regulate the metabolism of SCFAs and help to improve intestinal mucosal function, thereby attenuating the systemic inflammatory response by restructuring the gut microbiota.

The results of our KEGG pathway analysis showed that PROEs affected the tryptophan metabolic pathway, which plays an important role in the development of depression. The gut microbiota may affect neurotransmitters in the brain by regulating the metabolism of tryptophan and downstream metabolites such as kynurenic acid and quinolinic acid [47,48]. Studies have reported that mice that received fecal bacteria transplantation from chronic-stress mice had reduced adult hippocampal neurogenesis and damaged tryptophan metabolic pathways but that supplementing with 5-HTP, the direct precursor of 5-HT, alleviated depression-like behavior and restored hippocampal neurogenesis [49]. Tryptophan is an essential amino acid in the human body whose decomposition pathways include the 5-HT and kynurenine pathways.

Indoleamine 2,3-dioxygenase-1 (IDO1) is the key rate-limiting enzyme of tryptophan metabolism and is distributed in the intestine and brain. When IDO1 increases or inflammation induces hyperactivation, more tryptophan becomes metabolized into kynurenine, leading to tryptophan depletion and insufficient synthesis of 5-HT, which affects the function of both the cerebral cortex and hippocampus, inducing depression-like behaviors such as loss of interest and insomnia [50]. The ratio of kynurenine to tryptophan is therefore considered to be a sensitive indicator of IDO1 activity and cellular immune status [51]. Tryptophan–kynurenine metabolism has even been reported to be directly or indirectly regulated by gut microbiota [52]. Butyric acid, the main metabolite of gut microbiota, inhibits IDO1 expression in intestinal epithelial cells [53]. The systemic inflammatory response caused by an imbalance of gut microbiota can induce the inflammatory inducible enzyme IDO1, which affects the metabolism of tryptophan–kynurenine [54]. In addition, 5-HT_1A_ and 5-HT_2A_ receptors in the 5-HT system are most closely related to emotional disorders [55,56]. The 5-HT_1A_ receptor is the key target of 5-HT antianxiety drugs, which can inhibit anxiety-like behavior after activation [57,58]. The 5-HT_2A_ receptor antagonists have antianxiety effects, and 5-HT_2A_ knockout mice exhibited attenuated depression-like behavior [59,60]. In this study, PROEs restored the levels of tryptophan and kynurenine in plasma, increased the expression of 5-HT_1A_ mRNA in the cortex, reduced the expression of 5-HT_2A_ mRNA in the cortex, and reduced the expression of IDO1 mRNA in the cortex and duodenum of depression-model rats.

Gut microbiota are also closely related to the function of the HPA axis and neurotransmitter system. Hyperfunction of the HPA axis leads to increased secretion of cortisol, increased permeability of intestinal mucosa, changes in the composition of gut microbiota, and increased levels of LPS in serum, though transplantation of fecal bacteria from normal animals into depression-model animals can improve the HPA axis function of the recipients [61]. In addition, the gut microbiota can directly synthesize neurotransmitters such as 5-HT, NE, and DA, which may affect the function of central neurotransmitters through the circulatory system and vagus nerve [27,62]. Our Spearman correlation analysis showed that the levels of ACTH, CORT, and CRF in the HPA axis were negatively correlated with the relative abundance of gut microbiota protective bacteria and SCFA-producing bacteria (*Romboutsia*, *Roseburia*, and *Prevotella_9*) but were positively correlated with LPS-producing bacteria (*Bacteroides*). Additionally, the levels of 5-HT, DA, NE, and other neurotransmitters in the brain were positively correlated with the relative abundance of gut microbiota protective bacteria and SCFA-producing bacteria but were negatively correlated with LPS-producing bacteria. These results indicate that the improvements to the HPA axis and neurotransmitter function from PROEs may be related to their regulation of gut microbiota. However, the relationship between PROEs that antagonize the HPA axis and thereby enhance the function of neurotransmitter and gut microbiota still needs further study.

## 4. Materials and Methods

### 4.1. Reagents and Materials

PR was purchased from Beijing Renwei Chinese Medicine Decoction Pieces Co., Ltd., Beijing, China, and positively identified by Dr. Rong Luo, an associate professor at the School of Traditional Chinese Medicine in Capital Medical University (Beijing, China). Fluoxetine hydrochloride (FXT) as the positive drug was bought from Lilly Suzhou Pharmaceutical Co., Ltd. (Suzhou, China), and HPLC-grade acetonitrile and methanol were obtained from Fisher Scientific Inc. (Emerson, IA, USA). Rat adrenocorticotropic hormone (ACTH), corticotropin releasing factor (CRF), corticosterone (CORT)-releasing-factor, and lipopolysaccharide (LPS) ELISA kits were purchased from BLUE GENE (Shanghai, China). Interleukin-6 (IL-6) ELISA kits were purchased from NEOBIOSCIENCE (Beijing, China), and QIAamp DNA stool mini-kits were purchase from QIAGEN (Hilden, Germany).

Additionally, standards of norepinephrine (NE), dopamine (DA), 5-hydroxytryptamine (5-HT), 5-hydroxyindoleacetic acid (5-HIAA), high vanillic acid (HVA), dihydroxyphenyl acetic acid (DOPAC), acetic acid, propionic acid, butyric acid, 4-methylvaleric acid, and tryptophan (TRP) were purchased from Sigma-Aldrich Co. (St. Louis, MO, USA), and standards of kynurenine (KYN) were purchased from Yuanye Bio-Technology Co., Ltd. (Shanghai, China). Finally, the occludin monoclonal antibody was purchased from Proteintech Group, Inc. (Wuhan, China), and the β-actin antibody was purchased from Cell Signaling Technology (Beverly, CA, USA).

### 4.2. PROEs Preparation

The PR was refluxed three times with 8-fold 60% EtOH for 1.5 h, and then the solvent was removed to obtain the extracts of PR. These extracts further underwent column chromatography on MCI GEL CHP20P resin eluted with 20% EtOH, 50% EtOH, and 90% EtOH. The fraction of 50% EtOH was then gathered and concentrated to obtain the PROEs, resulting in an 8.4% extraction rate.

### 4.3. PROEs Ingredient Identification

A Waters Acquity UPLC system, coupled with a Q-TOF SYNAPT G2–Si high-definition mass spectrometer (Waters, Milford, MA, USA), was used to carry out LC-MS. First, the PROEs were separated on an Acquity UPLC BEH C18 column (1.7 μm, 2.1 × 100 mm) that was kept at 40 °C and at a flow rate of 0.4 mL/min. Acetonitrile (A) and 0.1% aqueous formic acid (B) were used as the mobile phase, and the gradient programs were as follows: 5–12% A (0–3 min), 12–20% A (3–6 min), 20–30% A (6–10 min), 30–35% A (10–10.5 min), 35–45% A (10.5–15 min), 45–50%A (15–18 min), and 50–95%A (18–20 min). The mass spectrometry instrument was equipped with an ESI-Ion source and was used to detect the spectrum in the *m*/*z* 50–2000 range with MS^n^ as the scan mode in negative mode. Here, the ion source temperature was 100 °C, and the desolvation gas temperature was 500 °C, and the flow rates of cone and desolvation gas were set at 50 L/h and 800 L/h, respectively. The voltages of the capillary and cone in negative ion mode were set at 2.5 kV and 40 V, respectively, and leucine enkephalin (*m*/*z* 554.2615 in negative ion mode) was used as a reference mass.

### 4.4. Animals and Treatments

All the animal studies in this paper were authorized by the Experimental Animal Management Committee of Capital Medical University (No: AEEI-2016-023). The 230 ± 20 g Sprague–Dawley rats (male, n = 56) were supplied by Beijing Vital River Laboratory Animal Technology Co., Ltd. (Beijing, China, SCXK (jing) 2016-0006). The rats were housed in a standard SPF environment at 24 ± 1 °C and 60 ± 5% humidity where a standard diet and water were freely available.

After acclimatization for one week, the rats were divided randomly into seven groups (each group = 8): normal control (NC), normal control + PROE (NC-PROE, 84 mg/kg), fluoxetine (FXT, 2 mg/kg), CUMS, CUMS + high-dose PROE (PROE-H, 126 mg/kg), CUMS + medium-dose PROE (PROE-M, 84 mg/kg), and CUMS + low-dose PROE (PROE-L, 42 mg/kg). The NC and NC-PROE groups were placed in a separate undisturbed area (three rats/cage), and the other groups (one rat/cage) were subjected to CUMS stressors according to methods previously described in the literature [63]. There were 9 stressors, namely, 4 °C cold-water swimming (5 min), food deprivation (24 h), water deprivation (24 h), tail clipping (1 min), flash stimulation (150 flashes per min, 24 h), white-noise exposure (24 h), binding (2 h), continuous illumination (24 h), and damp bedding (200 mL of water was put into 100 g of sawdust bedding for 24 h). One or two kinds of stimulation were randomly arranged every day for eight weeks. The NC and CUMS groups were given 5 mL/kg of distilled water, and the other groups were given corresponding drugs according to their body weights via intragastric administration for eight weeks, once a day. The weight of each rat was measured every two weeks.

### 4.5. Behavioral Tests

The rats’ behavioral characteristics were measured using the sucrose preference test (SPT), open-field test (OFT), and forced swimming test (FST), as determined according to methods described in the references [63].

### 4.6. ELISA Measurement

At the end of behavioral tests during the eighth week after the drug administrations, rats were anesthetized via 0.56% sodium pentobarbital, and blood was collected from the abdominal aorta for serum and plasma separation. Subsequently, the brain of each rat was immediately dissected on an ice plate and stored at −80 °C. The serum and hippocampus were then thawed at 4 °C, and the concentrations of CORT, CRF, ACTH, IL-6, and LPS in the serum were measured using ELISA kits following the manufacturer’s instructions.

### 4.7. Detection of Monoamine Neurotransmitters and Their Metabolites

In order to detect monamine neurotransmitters and their metabolites, the hippocampus from each rat was weighed and quickly homogenized in 120 μL of pretreatment solution A (0.4 mol/L of perchloric acid) on ice prior to being centrifuged at 12,000 rpm/min for 20 min at 4 °C after standing at room temperature for 30 min. Next, 90 μL of supernatant was collected, and 45 μL of pretreatment solution B (20 mmol/L of potassium citrate, 0.3 mol/L of dipotassium hydrogen phosphate, and 2 mmol/L of EDTA·2Na) was added. The sample was next vortexed and then centrifuged at 12,000 rpm/min for 20 min at 4 °C after also standing at room temperature for 30 min. The supernatants were collected and analyzed by high-performance liquid chromatography (HPLC) coupled with an electrochemical detector (Waters ECD2465, Milford, MA, USA) [64]. Here, the chromatographic column Waters symmetry shield RP 18 (150 × 3.9 mm, 5 μm, Waters Atlantis) was maintained at 30 °C, and the flow rate was 0.8 mL/min. The mobile phase (methanol–water, 8:92, *v*/*v*) was mixed with sodium acetate, 1-octanesulfonate, citric acid, and EDTA•2Na, and the detector potential was +0.6 V, with an injection volume of 40 μL.

### 4.8. Detection of Tryptophan and Kynurenine in Plasma

To detect the tryptophan and kynurenine, the rat plasma (100 μL) was mixed with 5% perchloric acid (100 μL) by vortex for 5 s in a centrifuge tube then centrifuged at 14,000 rpm/min for 10 min. The supernatants were then collected and analyzed by HPLC coupled with a UV detector (1200, Agilent, Santa Clara, CA, USA). The chromatographic column Diamonsil C18 (250 mm × 4.6 mm, 5 μm, DIKMA, Beijing, China) was maintained at 30 °C; the mobile phase A was 15 mmol/L sodium acetate buffer (pH 4.0), and mobile phase B was acetonitrile (92:8, *v*/*v*). The flow rate was 1.0 mL/min, with an injection volume of 50 μL, and the UV detection wavelengths of tryptophan and kynurenine were 280 and 360 nm, respectively.

### 4.9. Determination of Fecal Short-Chain Fatty Acids

The fecal concentrations of short-chain fatty acids (SCFAs), including acetic acid, propionic acid, and butyric acid, were analyzed by GC-MS (7000C, Agilent, Santa Clara, CA, USA), along with an HP-INNOWAX column (30 m × 0.32 mm × 0.25 μm, Agilent, Santa Clara, CA, USA). Standard solutions of acetic acid, propionic acid, and butyric acid were prepared at 500, 200, 100, 50, 20, 10, 5, 2, 1, and 0.5 μg/mL, respectively, and the internal standard substance of 4-methylvaleric acid was prepared at 10 μg/mL. Each fecal sample (80.0 mg) was soaked in 0.9 mL 0.5% phosphoric acid solution by vortexing and then centrifuging at 12,000 rpm/min for 5 min at 4 °C. Afterward, 700 μL liquid of supernatant was extracted with 800 μL of ethyl acetate and centrifuged at 12,000 rpm/min for 10 min at 4 °C. Next, 15 μL internal standard solution was added to 585 μL liquid of supernatant, which was vortexed for 15 sec and centrifuged at 12,000 rpm/min for 5 min. Finally, the supernatants were collected and measured by GC-MS.

### 4.10. Analysis of Intestinal Morphology

To analyze intestinal morphology, segments of the duodenum and colon from three rats in each group were fixed in freshly prepared 4% paraformaldehyde solution. These tissues were then dehydrated in a series of increasingly concentrated ethanol solutions, hyalinized in xylene, and embedded in paraffin wax. The samples were next cut into 5 μm sections and stained with hematoxylin and eosin (H&E). Finally, the sections were examined under a Panoramic SCAN (3DHistech, Budapest, Hungary), and the histopathology of duodenum and colon were analyzed with Image-Pro-Plus 6.0 image analyzer software (Media Cybernetics Inc., Rockville, MD, USA).

After the above analysis, segments of the duodenum and colon were collected and sectioned into pieces about 1 mm × 1 mm × 2 mm, fixed in 4 °C 2.5% glutaraldehyde buffer for 2 h, and washed with 0.1 M PB buffer 3 times for 15 min each time. These tissues were then fixed with 4 °C 1% osmic acid for 1 h, dehydrated using an ethanol gradient, embedded in epoxy resin, and sliced with an ultra-thin slicing machine at a thickness of 50 to 70 nm. Ultrastructural changes of the duodenum and colon epithelium were then observed under a transmission electron microscope (HT7700, HITACHI, Tokyo, Japan).

### 4.11. Western Blot Analysis

For Western blot analysis, colon tissues were cut into pieces and lysed in RIPA buffer with a protein phosphatase inhibitor, and the total proteins were extracted according to the manufacturer’s protocols. Then, the protein c concentration was detected by a BCA protein assay kit. Here, 50 μg of total protein was separated by 10% SDS-polyacrylamide gel electrophoresis and then transferred onto a 0.22 μm polyvinylidene fluoride membrane. After washing four times in 1× TBS buffer, the membranes were then blocked with 1× TBST containing 5% nonfat milk for 1 h at room temperature. Next, the membranes were incubated with primary antibodies overnight at 4 °C. The primary antibodies and their dilution concentrations were occludin (1:1000) and β-actin (1:1000). Subsequently, the membranes were washed with 1× TBST and incubated with secondary antibodies of goat anti-mouse or anti-rabbit IgG (1:20,000) for 1 h at room temperature. After the membranes were washed again using 1× TBST buffer, the proteins were visualized using enhanced chemiluminescence.

### 4.12. Quantitative Reverse-Transcription Polymerase Chain Reaction

Total RNA from the cerebral cortex and duodenum samples was extracted using RNA extraction reagent, and the RNA concentration was measured using the NanoDrop 2000 UV–vis spectrophotometer (Thermo, Waltham, MA, USA). The RNA was subsequently reverse transcribed to cDNA using a Servicebio^®^RT First Strand cDNA Synthesis Kit according to the manufacturer’s instructions. Amplification and quantitative detection were then performed in a real-time PCR machine (Bio-Rad, Hercules, CA, USA), where the RT–PCR protocol was as follows: initial denaturation at 95 °C for 10 min, then 40 cycles at 95 °C for 30 s and 60 °C for 30 s, and the 2^−ΔΔCt^ method was used for relative quantitative analysis of the results. Primer sequences for the genes of interest are listed in Table 2.

### 4.13. DNA Extraction and 16S rRNA Gene Sequencing of Fecal Samples

DNA from fecal samples was extracted using a QIAamp DNA stool mini-kit. The V3-V4 region of the bacterial 16S rRNA gene was then PCR amplified with primers 338F and 806R, and the sequencing library was prepared by a Truseq nano DNALT library prep kit (Illumina, San Diego, CA, USA). The effective sequences were merged and divided into operational taxonomic units (OTUs) with a 97% similarity cutoff via QIIME 2 software (version 2019.4), and the representative sequences of OTUs were then compared to the template sequences in the Greengenes database (Release 13.8) to be analyzed. Next, the fecal gut microbiome was analyzed by taxonomic composition analysis, alpha and beta diversity analysis, redundancy analysis, heatmap analysis, linear discriminant analysis of effect size, KEGG pathways, and PICRUSt analysis.

### 4.14. Statistical Analysis

GraphPad Prism 8.0.1, QIIME 2, and R software (v3.2.0) were used for all statistical analysis, and prior to this analysis, all data were expressed as mean ± standard error of the mean (SEM). One-way ANOVA was used for comparison between multiple groups, two-way ANOVA was used for changes in rat bodyweight, and the abundance of gut microbiota was compared between multiple groups using the Kruskal–Wallis rank-sum test.

## 5. Conclusions

PROEs play an antidepressant role by regulating the diversity and structure of gut microbiota, restoring intestinal barrier function, reducing the release of intestinal endotoxin, and lowering the level of inflammation. In addition, PROEs may also affect the level of neurotransmitters in the brain by regulating the tryptophan–kynurenine metabolic pathway to improve depression symptoms. However, further research is still necessary to study whether the antidepressant effects of PROEs depend on the existence of gut microbiota in combination with fecal bacteria transplantation in germ-free animals. Although the underlying effects of PROE on gut microbiota remain elusive, our present findings provide new insight and approaches for understanding the systemic mechanisms of the antidepressant effects of natural medicines and other products with good efficacy, poor absorption, and unclear mechanisms of action.

## Figures and Tables

**Figure 1 ijms-24-13877-f001:**
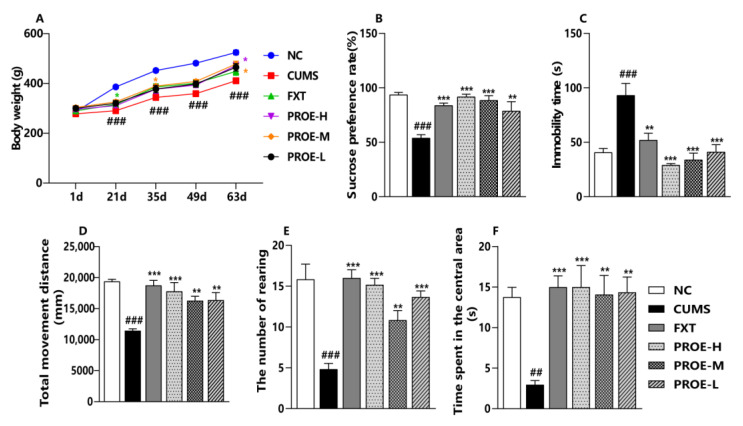
The effects of PROEs on CUMS-induced depression-like behaviors in rats. (**A**) The body weights of rats; (**B**) the sucrose preference rate; (**C**) the immobility time in the FST; (**D**–**F**) the total movement distance, the number of rearings, and the time spent in the central area of the OFT; data represent mean ± SEM (n = 8); ^##^ *p* < 0.01, and ^###^ *p* < 0.001 compared to the NC group; ** *p* < 0.01 and *** *p* < 0.001 compared to the CUMS group.

**Figure 2 ijms-24-13877-f002:**
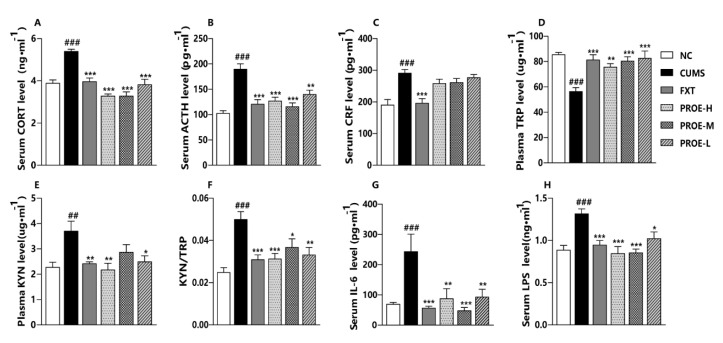
The effects of PROEs on HPA axis hormone, amino acid, and pro-inflammatory factors. (**A**–**C**) The levels of CORT, ACTH, and CRF in Serum; (**D**–**F**) the levels of TRP and KYN in plasma and the ratio of TRP/KYN; (**G**,**H**) the levels of IL-6 and LPS in serum; data represent mean ± SEM (n = 6); ^##^ *p* < 0.01 and ^###^ *p* < 0.001 compared to the NC group; * *p* < 0.05, ** *p* < 0.01 and *** *p* < 0.001 compared to the CUMS group.

**Figure 3 ijms-24-13877-f003:**
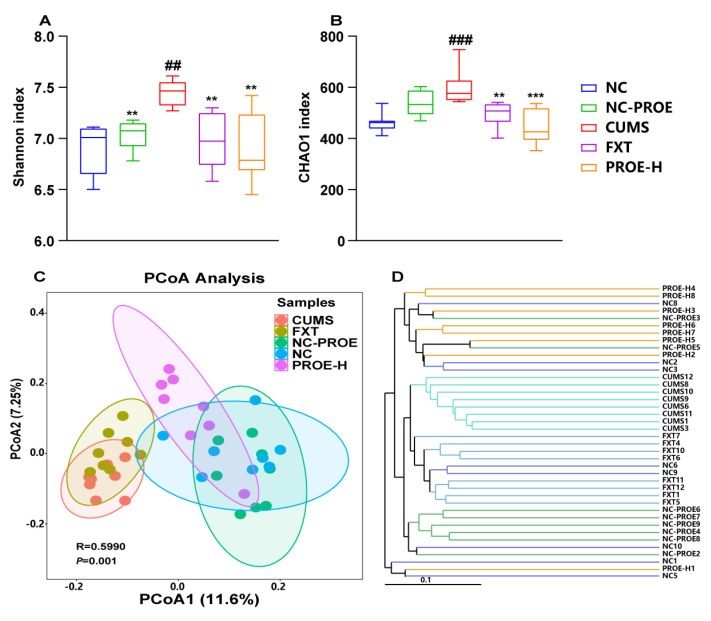
Changes in gut microbiota diversity. (**A**,**B**) Shannon index and Chao 1 index in alpha diversity analysis (n = 8); (**C**) principal co−ordinate analysis (PCoA); (**D**) unweighted pair group method with arithmetic mean (UPGMA); ^##^ *p* < 0.01 and ^###^ *p* < 0.001 compared to the NC group; ** *p* < 0.01 and *** *p* < 0.001 compared to the CUMS group.

**Figure 4 ijms-24-13877-f004:**
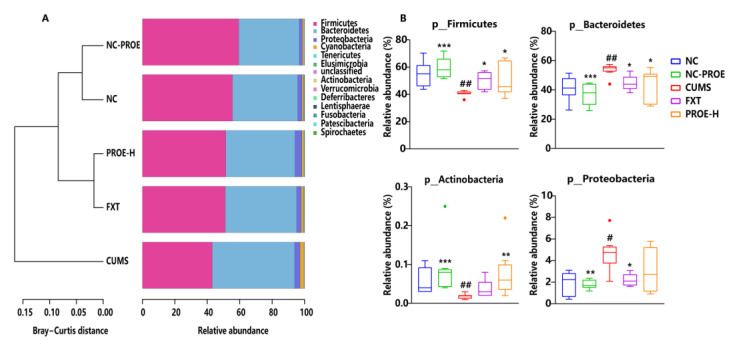
The effects of PROEs on the relative abundance of gut microbiota (**A**) and significantly changed gut microbiota (**B**) at the phylum level in depression-model rats (n = 8); ^#^
*p* < 0.05, ^##^ *p* < 0.01 compared to the NC group; * *p* < 0.05, ** *p* < 0.01 and *** *p* < 0.001 compared to the CUMS group.

**Figure 5 ijms-24-13877-f005:**
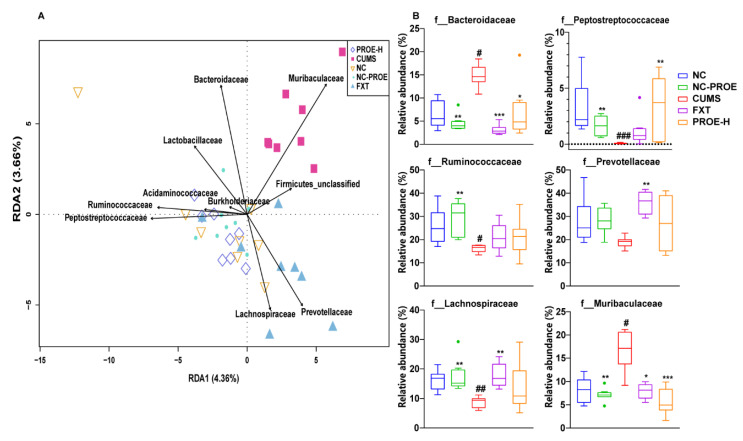
The effects of PROEs on the redundancy analysis (RDA) of gut microbiota (**A**) and significantly changed gut microbiota (**B**) at the family level in depression-model rats (n = 8); ^#^
*p* < 0.05, ^##^ *p* < 0.01 and ^###^ *p* < 0.001 compared to the NC group; * *p* < 0.05, ** *p* < 0.01 and *** *p* < 0.001 compared to the CUMS group.

**Figure 6 ijms-24-13877-f006:**
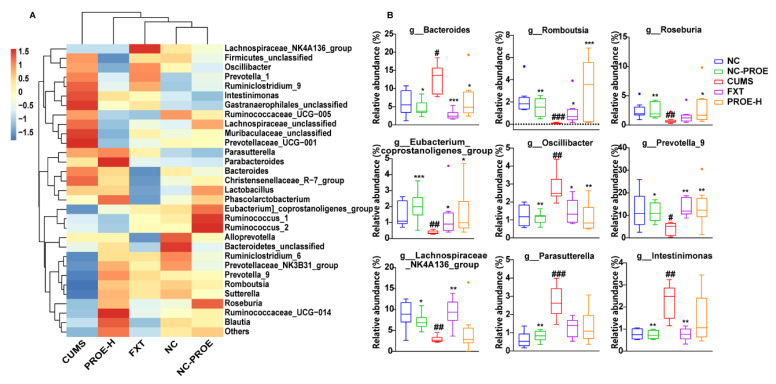
Heat map analysis of the effects of PROEs on gut microbiota (**A**) and significantly changed gut microbiota (**B**) at the genus level in depression-model rats (n = 8); ^#^
*p* < 0.05, ^##^ *p* < 0.01 and ^###^ *p* < 0.001 compared to the NC group; * *p* < 0.05, ** *p* < 0.01 and *** *p* < 0.001 compared to the CUMS group.

**Figure 7 ijms-24-13877-f007:**
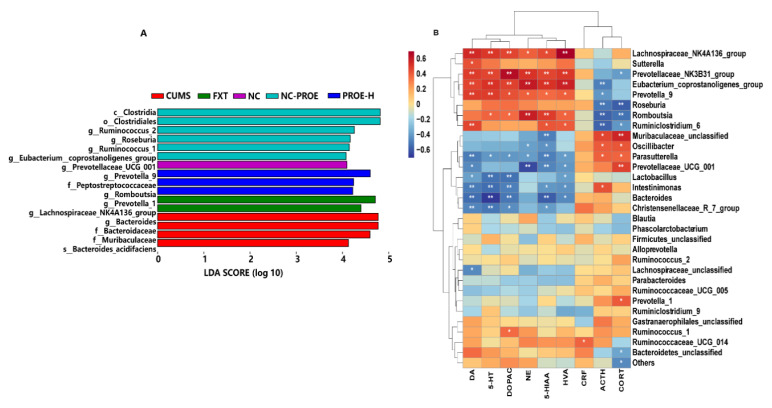
(**A**) Linear discriminant analysis of effect size (Lefse) with *p* < 0.01 and LDA score (log 10) > 4.0; (**B**) correlation analysis among brain neurotransmitters, HPA axis hormone, and relative abundances of gut microbiota. Red and green squares indicate negative and positive correlations, respectively, and the intensities of the colors are proportional to the degree of correlation. * *p* < 0.05, ** *p* < 0.01.

**Figure 8 ijms-24-13877-f008:**
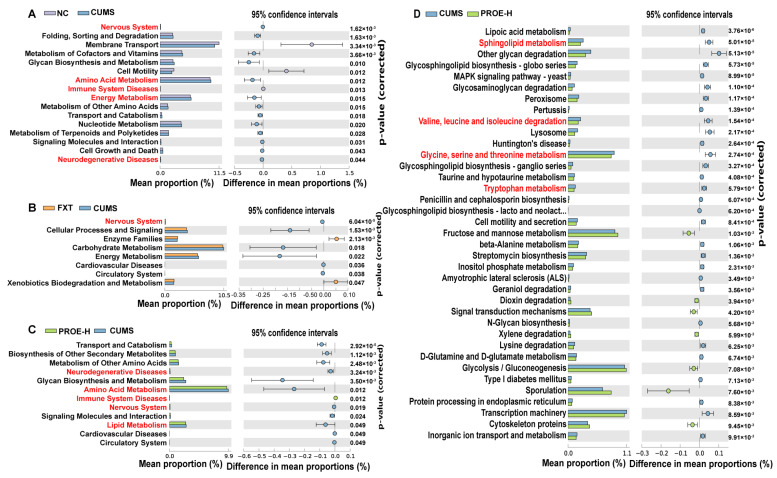
Predicted functions for the altered metagenome of gut microbiota in each group shown with KEGG pathways. (**A**–**C**) KEGG pathways at level 2; (**D**) KEGG pathways at level 3; the extended error bar plot of significantly differential KEGG pathways predicted using phylogenetic investigation of communities by reconstruction of unobserved states (PICRUSt). Only *p* < 0.05 are shown.

**Figure 9 ijms-24-13877-f009:**
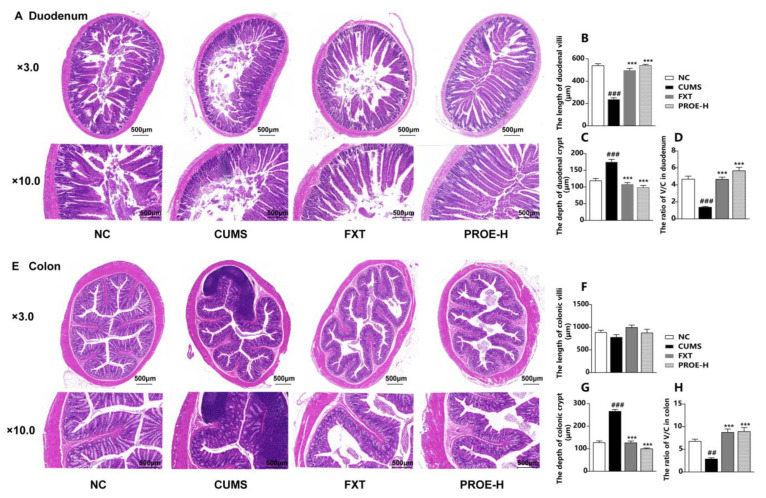
H&E staining of duodenum and colon tissue (n = 3). (**A**,**E**) The effects of PROEs on duodenum and colon histopathological changes; (**B**,**F**) the length of duodenal and colonic villi; (**C**,**G**) the depth of duodenal and colonic crypts; (**D**,**H**) the ratio of V/C in the duodenum and colon; data represent mean ± SEM (n = 10); ^##^ *p* < 0.01 and ^###^ *p* < 0.001 compared to the NC group; *** *p* < 0.001 compared to the CUMS group.

**Figure 10 ijms-24-13877-f010:**
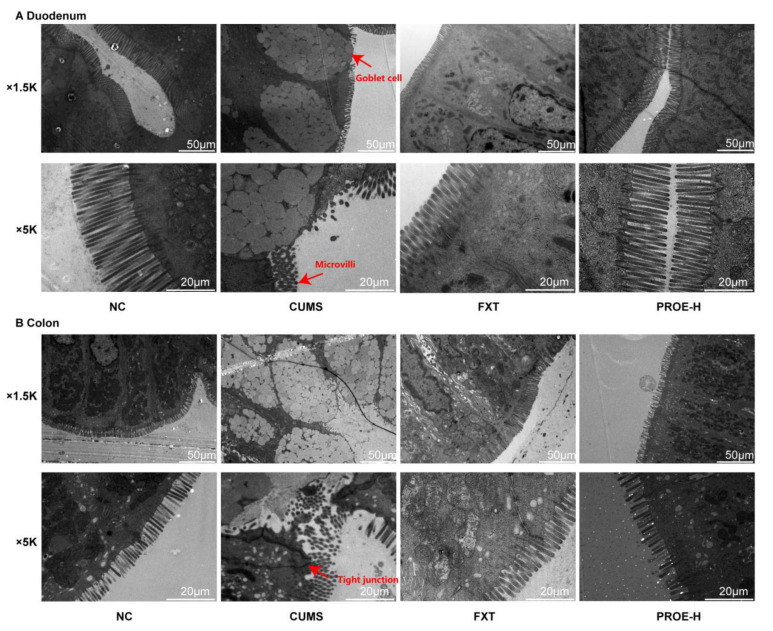
The effects of PROEs on duodenum (**A**) and colon (**B**) epithelium ultrastructures (n = 3) (scale bar = 50, 20 µm; original magnification ×1.5 k, ×5.0 k).

**Figure 11 ijms-24-13877-f011:**
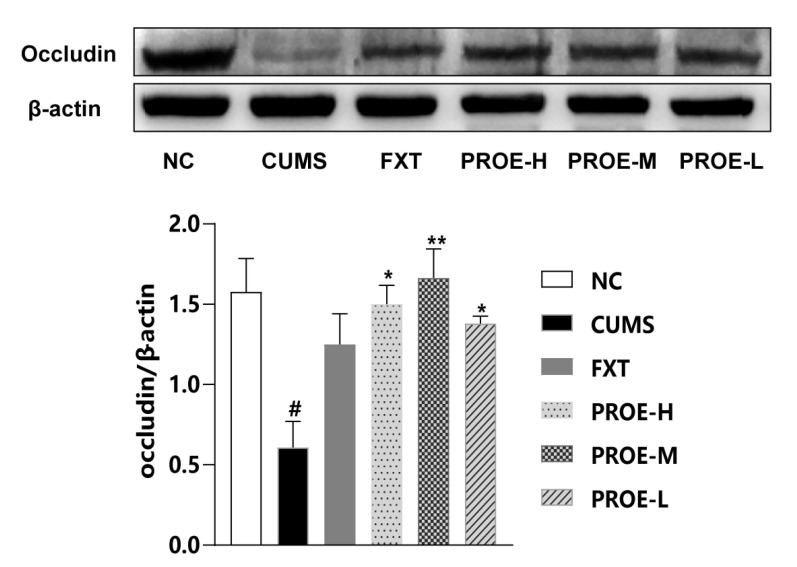
The effects of PROEs on the protein expression of occludin in the colon; data represent mean ± SEM (n = 3); ^#^
*p* < 0.05 compared to the NC group; * *p* < 0.05, ** *p* < 0.01 compared to the CUMS group.

**Figure 12 ijms-24-13877-f012:**
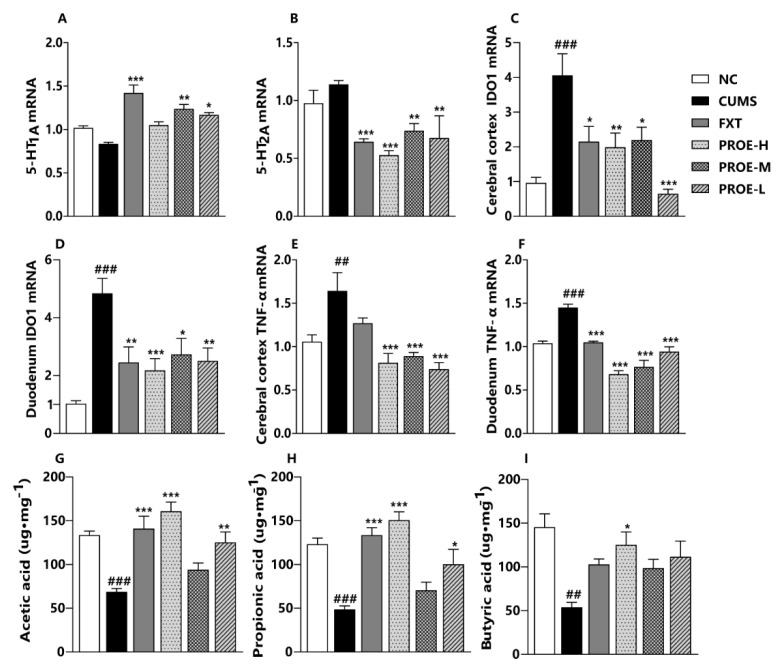
The effects of PROEs on the expression of 5-HT_1A_, 5-HT_2A_, IDO1, and TNF-α mRNA, and the concentrations of SCFAs. (**A**,**B**) The expression of 5-HT_1A_ and 5-HT_2A_ mRNA in the cerebral cortex; (**C**,**D**) the expression of IDO1 mRNA in the cerebral cortex and duodenum; (**E**,**F**) the expression of TNF-α mRNA in the cerebral cortex and duodenum; (**G**–**I**) the concentrations of acetic acid, propionic acid, and butyric acid in feces; data represent mean ± SEM (n = 6); ^##^ *p* < 0.01 and ^###^ *p* < 0.001 compared to the NC group; * *p* < 0.05, ** *p* < 0.01 and *** *p* < 0.001 compared to the CUMS group.

**Table 1 ijms-24-13877-t001:** The effects of PROEs on the monoamine neurotransmitters and their metabolites in the hippocampus of rats (mean ± SEM, n = 6).

Groups	NE (ng/g)	5-HT (ng/g)	DA (ng/g)	HVA (ng/g)	DOPAC (ng/g)	5-HIAA (ng/g)
NC	151.22 ± 17.07	110.66 ± 11.74	103.56 ± 11.99	101.95 ± 9.99	229.90 ± 17.71	215.39 ± 7.42
CUMS	58.02 ± 3.19 ^###^	18.52 ± 2.47 ^###^	14.97 ± 3.54 ^##^	15.91 ± 2.20 ^###^	52.85 ± 5.22 ^##^	113.96 ± 3.34 ^##^
FXT	180.78 ± 11.40 ***	109.55 ± 17.42 ***	97.13 ± 12.06 **	102.83 ± 7.07 ***	212.69 ± 15.70 **	206.62 ± 12.60 **
PROE-H	208.76 ± 13.46 ***	80.76 ± 14.46 **	98.03 ± 25.22 **	82.89 ± 21.88 **	215.91 ± 50.60 **	199.57 ± 17.74 *
PROE-M	122.08 ± 17.13 *	66.81 ± 6.30 *	74.18 ± 13.81	87.93 ± 9.51 **	206.01 ± 27.01 *	198.18 ± 14.91 *
PROE-L	168.83 ± 8.92 ***	81.51 ± 3.26 **	38.45 ± 16.21	62.67 ± 6.39	180.39 ± 35.47 *	195.92 ± 30.79 *

^##^ *p* < 0.01 and ^###^ *p* < 0.001 compared to the NC group; * *p* < 0.05, ** *p* < 0.01 and *** *p* < 0.001 compared to the CUMS group.

**Table 2 ijms-24-13877-t002:** Gene primer sequences.

Gene	Primer	Primer Sequence (5′ to 3′)	Product Size (bp)
*GAPDH*	Forward primer	5′CTGGAGAAACCTGCCAAGTATG3′	138
	Reverse primer	5′GGTGGAAGAATGGGAGTTGCT3′	
*TNF-α*	Forward primer	5′CCAGGTTCTCTTCAAGGGACAA3′	80
	Reverse primer	5′GGTATGAAATGGCAAATCGGCT3′	
*5-HT_1A_*	Forward primer	5′ACTTGGCTCATTGGCTTTCTCA3′	119
	Reverse primer	5′GAGTAGATGGTGTAGCCGTGGTC3′	
*5-HT_2A_*	Forward primer	5′TATGCTGCTGGGTTTCCTTGTC3′	201
	Reverse primer	5′TTGAAGCGGCTGTGGTGAAT3′	
*IDO1*	Forward primer	5′GATGAAGATGTGGGCTTTGCT3′	285
	Reverse primer	5′GCAGTAGGGAACGGCAAGA3′	

## Data Availability

The datasets generated and/or analyzed during the current study are included in this published article and the Appendix A.

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
