# Peer review of "Polygalae Radix Oligosaccharide Esters May Relieve Depressive-like Behavior in Rats with Chronic Unpredictable Mild Stress via Modulation of Gut Microbiota"

_ijms, 2023, doi:10.3390/ijms241813877_

Round 1
Reviewer 1 Report
Attached.

Author Response
Thank you very much for your valuable comments and suggestions on the paper. We have carefully revised the article based on your comments, and marked the changes in red font. The answers to each question are as follows:
Q1: In the Introduction section, I would like to see a paragraph about the neuroprotective effect of Polygalae Radix (PR) in the preclinical model and if it is beneficial in any other conditions. And also if PR is being used in clinical trials and safety profiles.
Reply: Thank you for your valuable suggestions. We have presented the neuroprotective role of PR in preclinical models and its clinical application in the introduction section.
Polygalae Radix (PR), the dried root of Polygala tenuifolia Willd. or Polygala sibiria L., is a well-known traditional Chinese medicine that has the effect of tranquilizing the mind, improving intelligence, and restoring normal coordination between the heart and kidneys, and this medicine is widely used for the treatment of insomnia, dreaminess, palpitations, and neurasthenia in clinic [10] in the form of Kai-Xin-San, Yuan-Zhi decoction, or Anshendingzhi decoction [11-13]. Modern pharmacological studies have also shown that shown that PR has beneficial antidepressant, sedative, anti-aging, neuroprotective, intelligence, and immune system effects [14]. Polygalae Radix oligosaccharide esters (PROEs) are the active ingredients in PR thought to be responsible for its antidepressive effects. Researchers have reported that 3,6'-dicalicosyl sucrose and Tenuifoliside A are the main components of PR total oligosaccharide esters and that these compounds regulate the level of monoamine neurotransmitters and improve the function of the HPA axis [15-17].
[10] Liu, P.; Hu, Y.; Guo, D.H.; Wang, D.X.; Tu, H.H.; Ma, L.; Xie, T.T.; Kong, L.Y. Potential antidepressant properties of Radix Polygalae (Yuan Zhi). Phytomedicine. 2010, 17, 794-9.
[11] Qu, S.; Liu, M.; Cao, C.; Wei, C.; Meng, X.E.; Lou, Q.; Wang, B.; Li, X.; She, Y.; Wang, Q.; et al. Chinese Medicine Formula Kai-Xin-San Ameliorates Neuronal Inflammation of CUMS-Induced Depression-like Mice and Reduces the Expressions of Inflammatory Factors via Inhibiting TLR4/IKK/NF-κB Pathways on BV2 Cells. Front. Pharmacol. 2021, 12:626949.
[12] Wu, Q.; Li, X.; Jiang, X.W.; Yao, D.; Zhou, L.J.; Xu, Z.H.; Wang, N.; Zhao, Q.C.; Zhang, Z. Yuan-Zhi decoction in the treatment of Alzheimer's disease: An integrated approach based on chemical profiling, network pharmacology, molecular docking and experimental evaluation. Front. Pharmacol. 2022, 13:893244.
[13] Yang, J.; Jiang, L.; Li, Y.; Zhu, Q.; Liu, D.; Xue, W.; Cao, B.; Liu, Y.; Wang, X. Clinical treatment of depressive patients with anshendingzhi decoction. J. Tradit. Chin. Med. 2014, 34(6):652-6.
[14] Zhao, X.; Cui, Y.; Wu, P.; Zhao, P.; Zhou, Q.; Zhang, Z.; Wang, Y.; Zhang, X. Polygalae Radix: A review of its traditional uses, phytochemistry, pharmacology, toxicology, and pharmacokinetics. Fitoterapia. 2020, 147:104759.
[15] Liu, X.; Wang, D.; Zhao, R.; Dong, X.; Hu, Y.; Liu, P. Synergistic Neuroprotective Effects of Two Herbal Ingredients via CREB-Dependent Pathway. Front. Pharmacol. 2016, 7:337.
[16] Hu, Y.; Liu, M.; Liu, P.; Guo, D.H.; Wei, R.B.; Rahman, K. Possible mechanism of the antidepressant effect of 3,6'-disinapoyl sucrose from Polygala tenuifolia Willd. J Pharm. Pharmacol. 2011, 63(6):869-74.
[17] Jin, Z.L.; Gao, N.; Zhang, J.R.; Li, X.R.; Chen, H.X.; Xiong, J.; Li, Y.F.; Tang, Y. The discovery of Yuanzhi-1, a triterpenoid saponin derived from the traditional Chinese medicine, has antidepressant-like activity. Prog. Neuropsychopharmacol. Biol. Psychiatry. 2014, 53, 9-14.
Q2: In the Results section, 2.2. The effect of PROEs on depression rat. In the second line, you should write full text for SPT, FST, and OFT since it is mentioned the first time. I had to go deep in the method section to find it.
Reply: In the Results section, 2.2, we wrote the full text for SPT, FST, and OFT, as shown below.
To evaluate the antidepressant activity of PROEs, we recorded the bodyweight of the rats and performed the sucrose preference test (SPT), open-field test (OFT), and forced swimming test (FST) on each group.
Q3: Authors should include the following paper also in their discussion. Radix Polygalae extract exerts antidepressant effects in behavioral despair mice and chronic restraint stress-induced rats probably by promoting autophagy and inhibiting neuroinflammation. Yunfeng Zhou a, Mingzhu Yan a, Ruile Pan a, Zhi Wang a, Xue Tao a, Chenchen Li a, Tianji Xia a, Xinmin Liu a b, Qi Chang a
Reply: We've added a reference to that article in the discussion section, as shown below.
In addition to mediating the microbiota, we also observed that PROEs relieved intestinal mucosal damage, increased the expression of occludin protein in the colon, adjusted the level of SCFAs in the feces and serum levels of LPS and IL-6, and reduced the expression of TNF-α m RNA in the cerebral cortex and duodenum. One study has reported that RP can inhibit the activation of NLRP3 inflammasome and the production of pro-inflammatory cytokines such as TNF-α in the prefrontal cortex of rats, thereby exerting antidepressant effects by promoting autophagy and inhibiting neuroinflammation [46].
[46] Zhou, Y.; Yan, M.; Pan, R.; Wang, Z.; Tao, X.; Li, C.; Xia, T.; Liu, X.; Chang, Q. Radix Polygalae extract exerts antidepressant effects in behavioral despair mice and chronic restraint stress-induced rats probably by promoting autophagy and inhibiting neuroinflammation. J. Ethnopharmacol. 2021, 265:113317.
Q4: Any particular reason for only using male rats in this study? If so, they should mention it in their title. I have noticed in the field, researchers sometimes only use male rats. Is it possible that only male rats have depressive-like behavior? Is it worth considering a similar study in female rats? If yes, please describe why female rats that were not part of this study. That means polygalae radix could only be used in males.
Reply: Before using this animal model, we have consulted relevant literatures. A study has conducted statistics on literature related to CUMS in CNKI and PubMed in the past 10 years [1]. The results show that 95.3% of the CUMS models selected male rats, 2.5% of the CUMS models selected females, and 2.2% of the CUMS models selected males and females. Another study reported that the majority (96.86%) of CUMS studies have been performed using adult male rodents (247 papers), Only 1 study included comparative analyses of both sexes, and 7 studies were conducted in female rodents [2]. It has been reported that male rats are more sensitive to chronic stress [3]. Therefore, we chose the male rat model doesn’t mean polygalae radix could only be used in males, but rather because the CUMS model is more suitable for using male rats. We will consider using a female rat model simultaneously in further study.
[1] Li, S.S.; Liu, X.M.; Wang, Q. Research on animal model of depression induced by chronic unpredictable mild stress based on literature database. Chin J Comp Med. 2021, 31(4): 77-83.
[2] Antoniuk, S.; Bijata, M.; Ponimaskin, E.; Wlodarczyk, J. Chronic unpredictable mild stress for modeling depression in rodents: Meta-analysis of model reliability. Neurosci Biobehav Rev. 2019, 99:101-116.
[3] Kikusui, T.; Nakamura, K.; Mori, Y. A review of the behavioral and neurochemical consequences of early weaning in rodents [ J]. Appl Anim Behav Sci. 2008, 110(1): 73-83.
Q5: It is not very clear how these compounds were administered in the rats. It just says, once a day.
Reply: In this study, these compounds were given by intragastric administration, as shown below.
The NC and CUMS groups were given 5 ml/kg of distilled water, and the other groups were given corresponding drugs according to their bodyweights via intragastric administration for eight weeks, once a day. The weight of each rat was measured every two weeks.
Thank you again for your review of the manuscript.

Reviewer 2 Report
In a study entitled "Polygalae radix oligosaccharide esters may relieve depressive-like behaviour in rats with chronic unpredictable mild stress via modulating gut microbiota,"
QiJun Chen and colleagues. It has been reported that Polygalae Radix (PR) is a well-known traditional Chinese medicine with antidepressive properties. PR is primarily composed of Polygalae Radix oligosaccharide esters (PROEs). The gut microbiota has been considered a key factor in depression. There is, however, no evidence that PROEs modulate gut microbiota to affect depression. In this study, we investigated the effect of PROEs on the gut microbiota of rats suffering from depression and the possible mechanisms involved. Solitary-rearing combined with chronic unpredictable mild stress (CUMS) induced depression in rats. The depression-like behavior, the influence on the hypothalamic-pituitary-adrenal (HPA) axis, the levels of monoamine neurotransmitters in the hippocampus and short-chain fatty acids (SCFAs) in feces were assessed, and serum lipopolysaccharide (LPS) and interleukin-6 (IL-6) levels were assessed using ELISA. Transmission electron microscopy was used to observe the ultrastructural changes in the duodenal and colonic epithelium. 16S rRNA sequencing was used to profile the gut microbiota. It was found that PROEs alleviate depression-like behavior in model rats, increase monoamine neurotransmitter levels in the brain, and reduce HPA axis hyperfunction. Furthermore, PROEs may be able to correct the imbalance of gut microbiota in model rats, decrease intestinal mucosal damage by increasing the relative abundance of gut microbiota with barrier protection functions, and alter levels of SCFAs in feces and LPS, IL-6 in serum. Regarding the present manuscript, I would like to make a few remarks.
-Please check the reference style according to the author's guidelines
-Why do the authors propose this Chinese medicine? Maybe the authors know a mechanism of action or even a proposed action in the microbiota.
- Microbiota should be introduced properly in the first section of the manuscript
-What was the depression model? This depression model is validated in other manuscripts
-The materials and methods section is summarized, and other researchers can read it to replicate the study. I request that you read this section again and add more information.
-Why was fluoxetine administered at a dose of 2 mg/kg?
-In order to gain a comprehensive understanding of the study, the section on animals and treatments needs to be improved
-Section 4.12, table 1 does not contain information about primers.
-A description of how PcoA analysis was performed, as well as the variables used to calculate it
-KEGG was calculated, but the information about it is not included in the M&M section.
Author Response
Thank you very much for your valuable comments and suggestions on the paper. We have carefully revised the article based on your comments, and marked the changes in red font. The answers to each question are as follows:
Q1: Please check the reference style according to the author's guidelines.
Reply: We have made corrections to the reference style in accordance with the author's guidelines, and the corrections have been marked in red in the manuscript.
Q2: Why do the authors propose this Chinese medicine? Maybe the authors know a mechanism of action or even a proposed action in the microbiota.
Reply: We propose Polygalae Radix (PR) because it is a well-known traditional Chinese medicine which widely used for the treatment of insomnia, sedation, antidepressant properties. Modern pharmacological studies have shown that PR has effects on antidepressant, sedative hypnosis, neuroprotective [10-17]. Polygalae Radix oligosaccharide esters (PROEs) are the active ingredients in PR responsible for its antidepressive effects, although studies have found that PROEs have poor absorption and low bioavailability, suggesting that it may exert an antidepressant effect via modulation of gut microbiota [18,19]. In this study, we built a rat depression model to evaluate the effects of PROEs on their gut microbiota and to investigate the possible antidepressant mechanisms of PROEs.
[10] Liu, P.; Hu, Y.; Guo, D.H.; Wang, D.X.; Tu, H.H.; Ma, L.; Xie, T.T.; Kong, L.Y. Potential antidepressant properties of Radix Polygalae (Yuan Zhi). Phytomedicine. 2010, 17, 794-9.
[11] Qu, S.; Liu, M.; Cao, C.; Wei, C.; Meng, X.E.; Lou, Q.; Wang, B.; Li, X.; She, Y.; Wang, Q.; et al. Chinese Medicine Formula Kai-Xin-San Ameliorates Neuronal Inflammation of CUMS-Induced Depression-like Mice and Reduces the Expressions of Inflammatory Factors via Inhibiting TLR4/IKK/NF-κB Pathways on BV2 Cells. Front. Pharmacol. 2021, 12:626949.
[12] Wu, Q.; Li, X.; Jiang, X.W.; Yao, D.; Zhou, L.J.; Xu, Z.H.; Wang, N.; Zhao, Q.C.; Zhang, Z. Yuan-Zhi decoction in the treatment of Alzheimer's disease: An integrated approach based on chemical profiling, network pharmacology, molecular docking and experimental evaluation. Front. Pharmacol. 2022, 13:893244.
[13] Yang, J.; Jiang, L.; Li, Y.; Zhu, Q.; Liu, D.; Xue, W.; Cao, B.; Liu, Y.; Wang, X. Clinical treatment of depressive patients with anshendingzhi decoction. J. Tradit. Chin. Med. 2014, 34(6):652-6.
[14] Zhao, X.; Cui, Y.; Wu, P.; Zhao, P.; Zhou, Q.; Zhang, Z.; Wang, Y.; Zhang, X. Polygalae Radix: A review of its traditional uses, phytochemistry, pharmacology, toxicology, and pharmacokinetics. Fitoterapia. 2020, 147:104759.
[15] Liu, X.; Wang, D.; Zhao, R.; Dong, X.; Hu, Y.; Liu, P. Synergistic Neuroprotective Effects of Two Herbal Ingredients via CREB-Dependent Pathway. Front. Pharmacol. 2016, 7:337.
[16] Hu, Y.; Liu, M.; Liu, P.; Guo, D.H.; Wei, R.B.; Rahman, K. Possible mechanism of the antidepressant effect of 3,6'-disinapoyl sucrose from Polygala tenuifolia Willd. J Pharm. Pharmacol. 2011, 63(6):869-74.
[17] Jin, Z.L.; Gao, N.; Zhang, J.R.; Li, X.R.; Chen, H.X.; Xiong, J.; Li, Y.F.; Tang, Y. The discovery of Yuanzhi-1, a triterpenoid saponin derived from the traditional Chinese medicine, has antidepressant-like activity. Prog. Neuropsychopharmacol. Biol. Psychiatry. 2014, 53, 9-14.
[18] Chen, Y.; Liu, X.; Pan, R.; Zhu, X.; Steinmetz, A.; Liao, Y.; Wang, N.; Peng. B.; Chang, Q. Intestinal transport of 3,6'-disinapoylsucrose, a major active component of Polygala tenuifolia, using Caco-2 cell monolayer and in situ rat intestinal perfusion models. Planta. Med. 2013, 79, 1434-9.
[19] Hu, Y.; Liu, P.; Guo, D.H.; Rahman, K.; Wang, D.X.; Xie, T.T. Antidepressant effects of the extract YZ-50 from Polygala tenuifolia in chronic mild stress treated rats and its possible mechanisms. Pharm. Biol. 2010, 48, 794-800.
Q3: Microbiota should be introduced properly in the first section of the manuscript.
Reply: We have given a proper introduction to the gut microbiota in the first part of the manuscript, as shown below.
The gut microbiome includes all the microorganisms that inhabit the gut and their respective genomes [4]. Gut microbiota regulate a variety of host metabolic pathways, including intestinal motility, intestinal barrier homeostasis, nutrient absorption, and fat distribution [5]. The interaction between gut microbiota and the intestine is also involved in the regulation of nervous system function, which affects the occurrence and development of various diseases, including both gastrointestinal diseases and psychiatric diseases [6].
[4] Cani, P.D. Human gut microbiome: hopes, threats and promises. Gut. 2018, 67(9):1716-1725.
[5] Bercik, P.; Verdu, E.F.; Foster, J.A.; Macri, J.; Potter, M.; Huang, X.; Malinowski, P.; Jackson, W.; Blennerhassett, P.; Neufeld, K.A.; et al. Chronic gastrointestinal inflammation induces anxiety-like behavior and alters central nervous system biochemistry in mice. Gastroenterology. 2010, 139(6):2102-2112.e1.
[6] Mayer, E.A.; Nance, K.; Chen, S. The Gut-Brain Axis. Annu. Rev. Med. 2022, 73:439-453.
Q4: What was the depression model? This depression model is validated in other manuscripts.
Reply: As a mood disorder disease, depression is characterized by anhedonia, unresponsiveness, poor appetite, hypokinesia, insomnia, and suicidal tendencies. At present, the most commonly used animal models of depression are mainly divided into stress models, transgenic animal depression models, drug-induced depression models, solitary or fractional models, and other models. The chronic unpredicted mild stress (CUMS) model is a common model for studying depression, and its theoretical basis is that animals will have symptoms such as tension, anxiety and depression when they are in an unsuitable environment, and long-term, chronic, unpredictable stimuli can accelerate the development of depression. The model mainly focuses on anhedonia, the core symptom of depression, and the stimuli used generally include: day and night reversal, food and water deprivation, cold-water swimming, crowding, tilting, flash stimulation, noise, etc. Generally, 1-2 random stimuli are applied every day, the animals are unpredictable, and the modeling time is 3-4 weeks. Finally, the behavioral tests such as sucrose preference test, open-field test, and forced swimming test were used for evaluation. After chronic unpredictable mild stress, animals have reduced physical exercise capacity, anhedonia and elevated plasma corticosteroids, etc., which are similar to human depressive symptoms, and can simulate some causes and symptoms of depressed patients more realistically, so it is an ideal and reliable animal model of depression [1,2]. A large number of studies have reported that the CUMS model has good surface validity (can induce anhedonia symptoms well) and predictive validity (behavioral changes can be reversed by multiple antidepressants), and its induced behavioral and neurobiochemical changes have good stability [3-6].
[1] Li, S.S.; Liu, X.M.; Wang, Q. Research on animal model of depression induced by chronic unpredictable mild stress based on literature database. Chin J Comp Med. 2021, 31(4): 77-83.
[2] Antoniuk, S.; Bijata, M.; Ponimaskin, E.; Wlodarczyk, J. Chronic unpredictable mild stress for modeling depression in rodents: Meta-analysis of model reliability. Neurosci Biobehav Rev. 2019, 99:101-116.
[3] Liu, P.; Song, S.; Yang, P.; Rao, X.; Wang, Y.; Bai, X. Aucubin improves chronic unpredictable mild stress-induced depressive behavior in mice via the GR/NF-κB/NLRP3 axis. Int Immunopharmacol. 2023, 123:110677.
[4] Mohammadi, S.; Naseri, M.; Faridi, N.; Zareie, P.; Zare, L.; Mirnajafi-Zadeh, J.; Bathaie, S.Z. Saffron carotenoids reversed the UCMS-induced depression and anxiety in rats: Behavioral and biochemical parameters, and hippocampal BDNF/ERK/CREB and NR2B signaling markers. Phytomedicine. 2023, 119:154989.
[5] Wang, Y.; Wang, X.; Wang, K.; Qi, J.; Zhang, Y.; Wang, X.; Zhang, L.; Zhou, Y.; Gu, L.; Yu, R.; et al. Chronic stress accelerates glioblastoma progression via DRD2/ERK/β-catenin axis and Dopamine/ERK/TH positive feedback loop. J Exp Clin Cancer Res. 2023, 42(1):161.
[6] Yu, H.; Chen, L.; Lei, H.; Pi, G.; Xiong, R.; Jiang, T.; Wu, D.; Sun, F.; Gao, Y.; Li, Y.; et al. Infralimbic medial prefrontal cortex signalling to calbindin 1 positive neurons in posterior basolateral amygdala suppresses anxiety- and depression-like behaviours. Nat Commun. 2022, 13(1):5462.
Q5: The materials and methods section is summarized, and other researchers can read it to replicate the study. I request that you read this section again and add more information.
Reply: We carefully read the materials and methods section, and made corrections and added more information, which is marked in red in the manuscript.
Q6: Why was fluoxetine administered at a dose of 2 mg/kg?
Reply: According to the instructions for Fluoxetine hydrochloride (Lilly Suzhou Pharmaceutical Co., Ltd Suzhou, China), the recommended dose for adults and elderly patients with depression is 20 mg per day. The conversion coefficient between humans and rats is about 7 according to the body surface area exchange algorithm of human and animal doses, and the dose of fluoxetine in experimental rats is 20 mg/70kg*7=2 mg/kg.
Q7: In order to gain a comprehensive understanding of the study, the section on animals and treatments needs to be improved
Reply: We have made improvements to the animal and treatment sections, as shown below.
There were 9 stressors including 4 ℃ cold-water swimming (5 min), food deprivation (24 h), water deprivation (24 h), tail clipping (1 min), flash stimulation (150 flashes per min, 24 h), white-noise exposure (24 h), binding (2 h), continuous illumination (24 h), and damp bedding (200 ml of water was put into 100 g of sawdust bedding for 24 h). One or two kinds of stimulation were randomly arranged every day for eight weeks. The NC and CUMS groups were given 5 ml/kg of distilled water, and the other groups were given corresponding drugs according to their bodyweights via intragastric administration for eight weeks, once a day. The weight of each rat was measured every two weeks.
Q8: Section 4.12, table 1 does not contain information about primers.
Reply: In section 4.12, Primer sequences for the genes of interest are listed in Table 2.
Table 2. Gene primer sequence
Gene |
Primer |
Primer sequence (5′to 3′) |
Product size(bp) |
GAPDH |
Forward primer |
5¢CTGGAGAAACCTGCCAAGTATG3¢ |
138 |
Reverse primer |
5¢GGTGGAAGAATGGGAGTTGCT3¢ |
||
TNF-α |
Forward primer |
5¢CCAGGTTCTCTTCAAGGGACAA3¢ |
80 |
Reverse primer |
5¢GGTATGAAATGGCAAATCGGCT3¢ |
||
5-HT1A |
Forward primer |
5¢ACTTGGCTCATTGGCTTTCTCA3¢ |
119 |
Reverse primer |
5¢GAGTAGATGGTGTAGCCGTGGTC3¢ |
||
5-HT2A |
Forward primer |
5¢TATGCTGCTGGGTTTCCTTGTC3¢ |
201 |
Reverse primer |
5¢TTGAAGCGGCTGTGGTGAAT3¢ |
||
IDO1 |
Forward primer |
5¢GATGAAGATGTGGGCTTTGCT3¢ |
285 |
Reverse primer |
5¢GCAGTAGGGAACGGCAAGA3¢ |
Q9: A description of how PcoA analysis was performed, as well as the variables used to calculate it.
Reply: PCoA analysis (Principal co-ordinates analysis) is a non-binding data dimensionality reduction analysis method, which can be used to study the similarity or difference in the composition of sample communities, similar to PCA analysis; The main difference is that PCA uses species (including OTU) abundance tables to map directly based on Euclidean distances, while PCoA is based on selected distance matrices, both of which identify potential principal components that affect differences in sample community composition through dimensionality reduction. PCoA analysis sort a series of feature values and feature vectors, and select the most important feature values in the top few and represent them in the coordinate system, the result is equivalent to a rotation of the distance matrix, which does not change the mutual position relationship between the sample points, but only changes the coordinate system. In this study, R (version 3.3.1) software was used for PCoA statistical analysis and graphing.
Q10: KEGG was calculated, but the information about it is not included in the M&M section.
Reply: In section 4.13, We added the analytical methods to describe the fecal gut microbiota of rats, as shown below.
The effective sequences were merged and divided into operational taxonomic units (OTUs) with a 97% similarity cutoff via QIIME software, and the representative se-quences of OTUs were then compared to the template sequences in the Greengenes database (Release 13.8) to be analyzed. Next, the fecal gut microbiome was analyzed by taxonomic composition analysis, alpha and beta diversity analysis, redundancy analy-sis, heatmap analysis, linear discriminant analysis of effect size, KEGG pathways, and PICRUSt analysis.
Thank you again for your review of the manuscript.

Round 2
Reviewer 2 Report
Thank you to the authors for taking into account my comments. Good review process. No further comments are required.